# Native Segmentation Vision Transformers

**Guillem Brasó    Aljoša Ošep    Laura Leal-Taixé**

NVIDIA

research.nvidia.com/labs/dvl/projects/native-segmentation

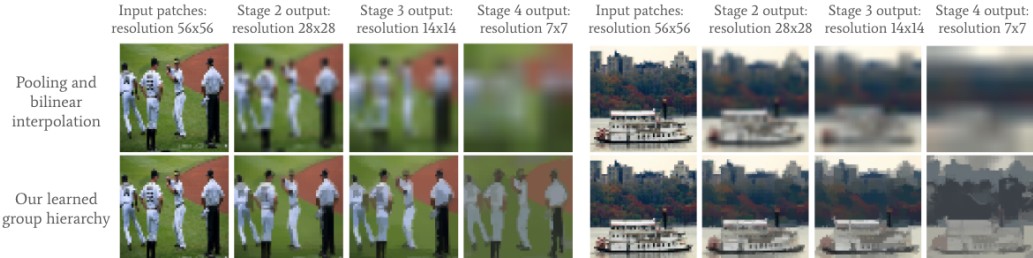

Figure 1: **Downsampling in vision backbones via uniform downsampling (*top*) *v.s.* learned downsampling (*bottom, this work*):** Vision backbones downsample feature maps using uniform-grid operators (*e.g.*, pooling, *top*) and rely on uniform upsampling (*e.g.*, bilinear interpolation, *top*) for image segmentation tasks. Our new backbone with spatial grouping layers learns to map pixels to a reduced set of tokens, aligning with image boundaries during downsampling (*bottom*). This enables scalable backbone-level *native* segmentation, *i.e.*, without the need for dedicated segmentation heads.

## Abstract

Uniform downsampling remains the de facto standard for reducing spatial resolution in vision backbones. In this work, we propose an alternative design built around a content-aware spatial grouping layer, that dynamically assigns tokens to a reduced set based on image boundaries and their semantic content. Stacking our grouping layer across consecutive backbone stages results in hierarchical segmentation that arises *natively* in the feature extraction process, resulting in our coined Native Segmentation Vision Transformer. We show that a careful design of our architecture enables the emergence of strong segmentation masks solely from grouping layers, that is, without additional segmentation-specific heads. This sets the foundation for a new paradigm of *native*, backbone-level segmentation, which enables strong zero-shot results without mask supervision, as well as a minimal and efficient standalone model design for downstream segmentation tasks.

## 1  Introduction

***Status quo.***  Modern hierarchical vision backbones [1, 2, 3] mirror the design principles of early convolutional networks [4], organizing feature processing across multiple stages at progressively lower spatial resolutions. While feature processing has been challenged, *e.g.*, convolutions *v.s.* self-attention, the downsampling stage has largely remained unchanged. Typically implemented via the ubiquitous *pooling* or, more recently, *strided convolutions* [1], these operations treat all spatial locations in a grid *uniformly*, irrespective of the image content. Such hierarchical feature extraction forms the foundation for state-of-the-art image segmentation methods, where dedicated segmentation heads [5, 6] learn to upsample and group the resulting features into semantically meaningful regions.

The uniform spatial treatment of features during downsampling manifests as feature misalignment during upsampling operations, placing an additional burden on decoder heads to compensate for inherent limitations in the backbone design [7, 8]. To this end, recent works [9, 10, 11, 12] explore

39th Conference on Neural Information Processing Systems (NeurIPS 2025).

alternative segmentation network designs and strategies for data-driven bottom-up pixel grouping based on their semantic content. Despite their conceptual appeal, these methods fall short against modern architectures due to either (i) algorithms with quadratic computational complexity relative to input resolution [9, 10], or (ii) non-differentiable grouping operations that limit their scalability and widespread practical use [11, 12], and necessitate the use of dedicated segmentation heads for downstream segmentation tasks, instead of capitalizing on their pixel-grouping capabilities.

**Native segmentation.** We introduce **Na**tive **Se**gmentation Vision **Tra**nsformer (SeNaTra), a backbone architecture whose core component, the ***spatial grouping layer***, replaces uniform grid-based downsampling with learned dynamic assignment of visual tokens to semantically coherent groups based on image content. Successive grouping operations across backbone stages naturally compose into a mapping from input pixels to final tokens, effectively creating a multi-scale hierarchy of segmentation masks for tokens at each backbone stage. We call this capability *native segmentation*, as it emerges from the backbone's inherent *region-aware* representation, rather than external heads [13, 6, 5]. It makes such external heads no longer strictly required, although empirically they can still be beneficial.

Our design has two main methodological advantages over prior backbone-level grouping work: (i) unlike methods using vanilla cross-attention [9, 10] or non-differentiable clustering [11, 12] we employ differentiable, iterative clustering inspired by perceptual grouping algorithms [14, 15], embedding a structured inductive bias that enables coherent groups to arise without direct supervision; (ii) we ensure scalability through *local* grouping layers with restricted context windows in early stages—enabling linear scaling with input resolution—while employing *dense* grouping only in the final stage to efficiently produce whole-image segmentation masks. Overall, our design enables scalable *native* segmentation while retaining efficiency and remaining end-to-end differentiable.

**Key findings.** We observe that in the absence of *any* mask supervision, super-pixel-like structures emerge as a consequence of our network design (Figure 1, *bottom*), akin to classical superpixel algorithms [16, 17, 18, 14], rather than being hand-crafted [19], or explicitly used as input [11]. These are further grouped into semantically meaningful regions in the final, dense grouping layer. We validate our native segmentation capability on zero-shot segmentation tasks across multiple established benchmarks and show that our model significantly outperforms prior art, including models trained on an order of magnitude larger datasets, suggesting our architecture is data-efficient, thanks to our grouping layer. When trained with explicit mask supervision for semantic and panoptic segmentation on ADE20k [20] and COCO-panoptic [21] our method outperforms several strong baselines *without* any dedicated segmentation heads, *e.g.*, RoI heads [5] or Transformer decoders [6], with a significantly reduced parameter and FLOP count. Furthermore, when used in conjunction with such heads, SeNaTra consistently improves the performance of top-performing backbones.

**In summary**, we (i) propose a Native Segmentation Vision Transformer, that learns a hierarchical segmentation of the visual input in the absence of any pixel/mask supervision. The key building block of our network is (ii) our grouping layer that performs image-content-adaptive feature downsampling, effectively replacing uniform, grid-based feature down/up-sampling layers, employed in consolidated segmentation networks. Finally, (iii) we unveil a streamlined native segmentation network that obtains masks in the absence of any dedicated heads, and excels at zero-shot segmentation, trained without any pixel/mask supervision, as well as on standard semantic/panoptic segmentation benchmarks.

## 2   Related Work

**Vision backbones** Since the pioneering work of Neocognitron [22] and LeNet [23], Convolutional Neural Networks (CNNs) have been propelling the advancements in data-driven computer vision. These networks typically employ a hierarchy of convolutional layers that apply a set of learnable filters to the input feature map, which are alternated with feature downsampling operations, yielding *hierarchy* of multi-scale feature maps. Despite the rise of plain transformer-based architectures [24], modern hierarchical backbones [1, 3, 25] are still dominant in dense prediction [26] and still adhere to the same underlying design principle: they are organized in multiple feature extraction stages, with uniform downsampling operations among them. In this work, we put our focus on the largely overlooked downsampling operation, and show that by replacing it with our proposed spatial grouping module, we can obtain a backbone with *native segmentation* capabilities.

**Dense prediction.** In the last decade, we witnessed a Cambrian explosion in network design for dense prediction. Notable examples include Fully Convolutional Networks [27], encoder-decoder architectures [28], and the pioneering work of [29, 30]. More recently, DETR [31] tackled end-to-end detection as set prediction using Transformers, treating object proposals or segments as learnable *queries*. MaskFormer [13, 6] capitalized on this design, and added a pixel decoder to upsample feature maps, and trained it jointly with a backbone and transformer decoder to process queries. SeNaTra can be used in conjunction with such segmentation heads to improve segmentation accuracy, or produce high-quality *native* masks in the absence of such dedicated heads. Recent large-scale efforts such as SAM and SAM2 [32, 26] have focused on introducing promptable segmentation and scaling, combining massive data generation pipelines with standard ViT-based backbones [24, 33]. In contrast, our work targets architectural innovation at the backbone level, introducing native segmentation capabilities with potential to complement or simplify such frameworks.

**Perceptual grouping.** Prior to the advent of end-to-end segmentation methods, combinatorial optimization was the main algorithmic tool for this task. Notable examples include the seminal work of [17], which introduced efficient graph-based segmentation to adaptively merge regions based on internal variation, and normalized cuts [34]. Traditional superpixel algorithms, such as SLIC [14], emerged as efficient tools to obtain segments based on color similarity and proximity. Recognizing segmentation's inherent ambiguity, several methods explored progressively merging regions into hierarchies of segments across multiple scales [18, 35]. Our approach draws inspiration from these but reformulates them in the context of modern, end-to-end trainable vision backbones.

Several methods proposed *learning-based* mechanisms for pixel grouping. [36] introduced a differentiable variant of the SLIC algorithm for task-specific superpixels. Similarly, [15] proposed a differentiable variant of K-Means for unsupervised object discovery, which iteratively assigns image pixels to a set of slots. While these approaches inspire our spatial grouping layer, we instead propose a sparse and efficient design, and integrate it as a fundamental building block for modern backbones.

**Grouping in vision backbones.** GroupViT [9] and ClusterFormer [10] pioneered the design of data-driven backbones with learnable downsampling operations. They group image constituents into a reduced set of tokens using (dense) cross-attention layers, which hinder their scalability due to the quadratic complexity of the attention operation *w.r.t.* input size. An orthogonal line of work [37] accelerates vision transformers for classification tasks with heuristic-based token merging strategies. By contrast, our approach is general, fully learnable, and scalable to large input resolutions as early *local* layers rand educe input token set cardinality on which dense layers operate. This enables our approach to be used in a variety of segmentation tasks and also significantly outperforms cross-attention-based grouping [10] in text-supervised semantic segmentation. Alternatively, [11, 12] mitigate this issue using non-differentiable super-pixel method [19] to obtain initial image segmentation followed by data-driven grouping, while TCFormer [12] relies on an external clustering method to group image constituents across multiple network layers. Our approach does not require such non-differentiable clustering methods and consists solely of differentiable grouping layers. Our streamlined design performs favorably compared to prior art in zero-shot segmentation. Moreover, unlike the aforementioned works, we show it performs favorably both with and *without* dedicated segmentation heads in downstream segmentation tasks.

## 3 Native Segmentation Vision Transformers

Our Native Segmentation Vision Transformer (SeNaTra) follows the standard structure of modern hierarchical vision backbones [1, 2, 3], consisting of four stages that progressively reduce the spatial resolution of feature maps while doubling their channel dimensions (Figure 2). Given an input image of size $H \times W$, the initial stage splits it into $4 \times 4$ patches to obtain initial token embeddings, and each subsequent stage $S_i$, $i = 2, \ldots, 4$ produces tokens at a resolution of $(H/2^{i+1}) \times (W/2^{i+1})$.

In Section 3.1, we describe our spatial grouping layer that replaces uniform downsampling layers in-between network stages. By composing these grouping layers our backbone builds a hierarchical image representation that organizes pixels into increasingly large, semantically meaningful regions (Figure 2 (a)). While our approach is general and task-agnostic, our learned downsampling operation further enables boundary-preserving feature upsampling, especially beneficial in downstream dense prediction tasks, such as segmentation, as presented in Section 3.2.

## 3.1 Content-aware Spatial Grouping Layer

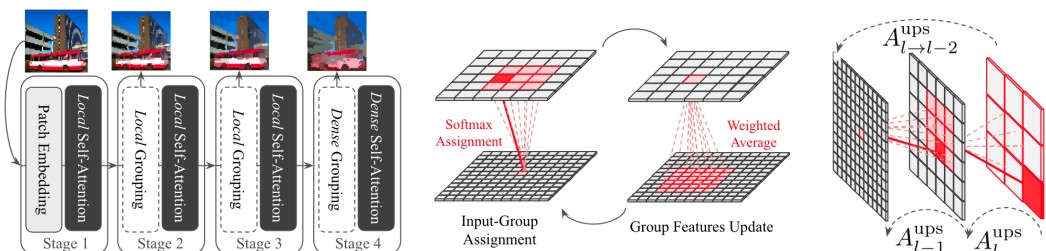

(a) **Architecture overview**    (b) **Spatial Grouping Layer**    (c) **Learned upsampling**

Figure 2: **Overall model design.** Visualization of our hierarchical architecture and its key components. (a) Our backbone architecture consists of four processing stages interconnected by content-aware grouping layers for downsampling. (b) Core operations of our Spatial Grouping Layer, which computes soft token assignments and updates group features iteratively (detailed in Algorithm 1). (c) The composition of learned assignment matrices across grouping layers in consecutive backbone stages enables principled feature upsampling.

**Learning semantically meaningful pixel groups.** *Uniform* downsampling operations such as pooling or strided convolutions, which are *de facto* standard in current architectures, treat all feature locations in an image equally regardless of their feature content, and apply a fixed operation for all input tokens. This approach is inherently limited in its ability to distinguish between high and low-frequency regions and capture relevant details. To address this limitation, we propose to *learn* a mapping between input and downsampled tokens that dynamically adapts to input features, instead of solely relying on feature positions in a grid. Specifically, we map tokens with similar feature embeddings, hence belonging to the same object or semantically meaningful region, to the same output token in our downsampled representation. By learning such mapping, our model preserves semantically meaningful boundaries within the image across its consecutive network stages.

**Grouping algorithm.** Building on this intuition, we frame our task as a differentiable clustering process inspired by the *K-means* [38, 39] and its modern differentiable variant [15] where our output downsampled tokens act as centroids, and input tokens are iteratively assigned to them. Formally, let $X^{\text{in}} \in \mathbb{R}^{N^{\text{in}} \times d}$ denote a set of $N^{\text{in}}$ $d$-dimensional input tokens, which correspond to either pixel embeddings or tokens from a previous stage. We aim to produce a reduced set of $N^{\text{out}}$ $2d$-dimensional tokens with reduced spatial dimensions. Following standard architecture designs, we set $N^{\text{out}} = N^{\text{in}}/4$ for all layers.

Our full approach is outlined in Algorithm 1. We first initialize $X^{\text{out}}$ with a strided convolution, as it is common practice [25, 3]. Then, for $L$ iterations ($L = 3$

---

**Algorithm 1 Grouping layer over an input feature map $X$ for $L$ iterations with sparsity**.

**Input:** Feature map $X^{\text{in}} \in \mathbb{R}^{N^{\text{in}} \times d}$, Mask[1] $M_{\text{loc}} \in \{0,1\}^{N^{\text{in}} \times N^{\text{out}}}$

**Learnable Modules:** Strided Conv `Conv`; linear projections $Q, K, V$; `MLP`, `LN`; rel. pos. bias $B$; temp. $\tau$.

1: $X^{\text{out}} \leftarrow \text{LN}(\text{Conv}(X))$
2: **for** $l = 1, \ldots, L$ **do**
3:     $A \leftarrow \tau k(X^{\text{in}}) \times q(X^{\text{out}})^T + B$
4:     $A \leftarrow A + M_{\text{loc}}$
5:     $A^{\text{ups}} \leftarrow \text{softmax}_{\text{rows}}(A)$
6:     **for** $i = 1, \ldots, N^{\text{in}}, j = 1, \ldots, N^{\text{out}}$ **do**
7:         $A^{\text{down}}_{ij} \leftarrow \dfrac{A^{\text{ups}}_{ij}}{\sum_{k=1}^{N^{\text{in}}} A^{\text{ups}}_{kj}}$
8:     **end for**
9:     $X^{\text{out}} \leftarrow X^{\text{out}} + \text{LN}((A^{\text{down}})^T \times v(X^{\text{in}}))$
10:     $X^{\text{out}} \leftarrow X^{\text{out}} + \text{LN}(\text{MLP}(X^{\text{out}}))$
11: **end for**
12: **return** $X^{\text{out}}, A^{\text{down}}, A^{\text{ups}}$

---

in our experiments), we alternate between two key steps: (i) computing a soft assignment matrix from input tokens with a cross-attention-like operation (L3-5), and (ii) renormalizing this matrix over columns to update $X^{\text{out}}$ with a weighted mean over input tokens (L6-9). Intuitively, since $A^{\text{ups}} \in [0,1]^{N \times N^{\text{down}}}$ is row-normalized, each element $A^{\text{ups}}_{ij}$ can be interpreted as the probability that each input token $X^{\text{in}}_i$ gets mapped to an output downsampled token $X^{\text{out}}_j$. These assignment probabilities are then used to update the corresponding features of $X^{\text{out}}$ (L9), which act as centroids.

---

[1]For clarity, we show a naive implementation with sparsity via $M_{\text{loc}}$(0 for enabled pairs, $-\infty$ otherwise). See Appendix E.1 for our efficient implementation.

By repeating this process over $L$ steps, we iteratively refine both the assignment probabilities as well as the resulting features $X^{\text{out}}$.

**Local and dense grouping.** A key limitation of Algorithm 1 lies in the cost of computing $A^{\text{ups}}$ (L3) due to the quadratic complexity *w.r.t.* the cardinality of the input token set, $N^{\text{in}}$, making it impractical for high-resolution feature maps. Inspired by the SLIC algorithm for superpixel generation [14, 36], for high-resolution feature maps we restrict the computation of cross-attention coefficients to a small $3 \times 3$ local window centered around each output token in $X^{\text{out}}$ (see Figure 2 b). Intuitively, this mechanism retains the flexibility of a learned downsampling operator, where input tokens can be dynamically mapped to their downsampled counterparts, and injects a locality prior: input tokens will be mapped to tokens that will be *close* in the resulting output space. This enables the notion of locality in output tokens, allowing us to leverage commonly used local attention mechanisms [1, 3]. Computationally, this prior results in highly sparse $A^{\text{ups}}$ and $A^{\text{down}}$ matrices that can be efficiently computed with CUDA kernels (see Appendix E.1) and, overall, reduces the computational complexity of our *Grouping Layer* from $\mathcal{O}(LN^2d)$ down to $\mathcal{O}(LNd)$ making it practical for high-resolution maps. In our architecture, we use local grouping in the second and third stages, where higher-resolution feature maps are processed. In the final stage, we enable *dense*, *i.e.*, non-sparse, grouping, which ensures that our model's output tokens can merge regions and objects over the entire input image.

**Connection to Slot Attention.** The core operations in our grouping layer are akin to those introduced in Slot Attention [15]. Our downsampled tokens can be interpreted as *slots* that, instead of being sampled from a random distribution, are initialized by a strided convolution layer over input tokens $X$. Additional technical differences include replacing the GRU originally used to update *slots*, *i.e.*, pixel groups, with a simpler skip connections (Algorithm 1, L9) and the use of *relative* positional encodings to encode spatial relationships between input and output tokens (Algorithm 1, L3). More importantly, the sparsity constraints in the cross-attention operation introduced in the previous paragraph enable efficient processing of high-resolution inputs, making this differentiable grouping mechanism practical for hierarchical vision backbones.

## 3.2 Native Segmentation

**Composing assignments via Markov chain.** By forwarding an image through our model, the combined output of all $n$ grouping layers yields two sets of matrices $\{A_i^{\text{ups}}\}_{i=1}^n$, and $\{A_i^{\text{down}}\}_{i=1}^n$, where each matrix $A_i^{\text{ups}}$ (resp. $A_i^{\text{down}}$) corresponds to the output of the grouping layer at stage $i+1$, with dimensions $N_i^{\text{in}} \times N_i^{\text{out}}$. As grouping layers are applied in consecutive stages, $N_i^{\text{out}} = N_{i+1}^{\text{in}}$ for each $i = 1, \ldots, n-1$. Now, recall that by construction, $A_i^{\text{ups}}$ is a row-stochastic matrix, where entries can be interpreted as the probability of each input token being mapped to a subsequent downsampled token. Each matrix $A_i^{\text{ups}}$ can therefore be interpreted as a *state transition matrix*, and the overall mapping from tokens at stage $l$ to tokens at an *earlier* stage $l-k \in \{1, \ldots n-1\}$ can be interpreted as a Markov chain with state transition probabilities given by:

$$A_{l \to l-k}^{\text{ups}} := A_{l-k+1}^{\text{ups}} \times \cdots \times A_l^{\text{ups}}, \qquad A_{l \to l+k}^{\text{down}} := (A_{l+k-1}^{\text{down}})^T \times \cdots \times (A_l^{\text{down}})^T. \qquad (1)$$

Where, analogously, since $A^{\text{down}}$ is a *column*-stochastic matrix, $A_{l \to l+k}^{\text{down}}$ defines a mapping for tokens from stage $l$ to $l+k$. Therefore, any set $X$ of arbitrary of $N_l^{\text{out}}$ $d$-dimensional token embeddings at stage $l$ can be upsampled to stage $l-k$ (resp. downsampled to $l+k$) resolution via dot product $A_{l \to l-k}^{\text{ups}}X$, (resp. $A_{l \to l+k}^{\text{down}}X$). Since all except for our last grouping layers utilize *local* grouping, at most one matrix in the product of assignment matrices will be non-sparse. The product of all sparse matrices involved is also block-sparse and can be efficiently computed (see Appendix E.1).

**Backbone-level segmentation.** The observations made in the previous paragraph enable a *probabilistic* interpretation of hierarchically decomposing an image into segments. At each stage $i$, $A_{1 \to i}^{\text{ups}}$ maps input tokens, *i.e.*, image patches, to $N_i^{\text{out}}$ disjoint tokens, *i.e.*, segments, where $N_i^{\text{out}}$ decreases with $i$. Our final stage 4 enables *dense* grouping, allowing tokens to encode segmentation masks spanning the entire image. Notably, this can be achieved without explicit supervision of intermediate transition matrices or their composition. Since grouping layers are differentiable, our entire architecture remains end-to-end trainable on standard image-level objectives through global pooling of final-stage tokens. At inference time, applying a learned classification head or text embeddings to final tokens, followed by upsampling via $A_{1 \to n}^{\text{ups}}$, enables *zero-shot* input-level predictions suitable

for semantic segmentation. Despite the absence of mask supervision, our grouping layer's strong inductive bias yields high-quality masks in this setup, as we show in Section 4.1.

**Leveraging mask supervision.** Image segmentation tasks can be divided into partitioning an image into $S$ disjoint segments, and doing *per-segment* classification. While contemporary methods rely on specialized heads to enable *instance-level* high-resolution predictions [6, 40], our model *directly* encodes image partitions through input-output token mappings $A_{1 \rightarrow n}^{\text{ups}}$ at the backbone level. This enables a minimalistic *purely native* approach: training only MLPs to classify our final tokens with bipartite-matching losses. Furthermore, our model can be integrated into standard segmentation frameworks with a key improvement: feature map upsampling and downsampling operations, commonly used in pixel decoders, can be replaced with our grouping-based operations, leading to improvements over the segmentation accuracy of state-of-the-art methods (Section 4.2).

# 4 Experiments

**Overview.** In the following, we extensively evaluate SeNaTra with *w.r.t.* different supervision regimes and task complexity. In Section 4.2, we start with *mask-free* supervision and study emerging segmentation from image-class (Section 4.1.1) and image-caption (Section 4.1.2) supervision, comparing our model to state-of-the-art zero-shot segmentation methods. In Section 4.2, we train and evaluate our model on standard datasets and benchmarks for semantic (Section 4.2.1) and panoptic (Section 4.2.2) segmentation, comparing our direct segmentation model and backbone as drop-in replacement against state-of-the-art. We analyze our design choices and contributions in Section 4.3.

**Models.** We evaluate three SeNaTra models: *tiny* (T), *base* (B), and *large* (L), with output embedding dimension of 512, 1024, and 1536, following [3]. Full configurations are provided in Appendix D.

## 4.1 Learning Without Mask Supervision

### 4.1.1 ImageNet Classification

We train SeNaTra on ImageNet-1k and ImageNet-22k [41], following the training setup of [1]. We visualize the learned group representations at different backbone stages in Figure 3, along with final per-group activations for the predicted class ($5^{th}$ col), and refer to Appendix D.1 for quantitative analysis and comparison with standard backbones [3]. While our network performs on-par with state-of-the-art on ImageNet classification task, we observe that as a by-product of our network design, our network produces a hierarchy of boundary-preserving *super-pixel-like* groups, combined in the last, *dense* grouping layer into meaningful semantic regions ($4^{th}$ col). We emphasize we train our models using *output-level* class supervision only. *Our model retains state-of-the-art performance w.r.t. classification, and, remarkably, learns per-pixel localization of objects without mask supervision as a direct consequence of our proposed architectural changes.*

### 4.1.2 Zero-shot Segmentation from Vision-Language Supervision

**Setup.** We pre-train SeNaTra with image-text pairs using softmax contrastive objective [42, 43], borrowing hyperparameters from [44]. We evaluate our models in zero-shot semantic segmentation. To obtain image group embeddings, we apply a linear projection layer to the final image (resp. text) output tokens and apply global pooling, followed by $L2$ normalization. To classify, we feed class names (for each dataset) through the text encoder with standard template prompts and pick the class with maximum cosine similarity for each group embedding, followed by our upsampling operation (Section 3.2). For details, see Appendix D.2.

**Datasets.** Following [44], we train our model for 20 epochs from scratch on the union of the CC3M [45] and CC12M [46] datasets (20M semi-curated image-text pairs), and union including Red-Caps12M dataset [47] (+12M additional pairs). Following [48], we evaluate trained models on Pascal VOC [49], Pascal Context [50], COCO [51], COCO-Stuff [52], ADE20k [20] and Cityscapes [53]. These span diverse scenarios, ranging from urban street scenes (Cityscapes), general object categories (COCO, Pascal VOC), and densely annotated fine-grained scenes (ADE20k, Pascal Context). We discuss results in terms of standard mean intersection-over-union (mIoU).

**Discussion.** As can be seen in Table 1, our SeNaTra outperforms specialized state-of-the-art methods across most benchmarks, including models leveraging CLIP's large-scale pre-training on 400M

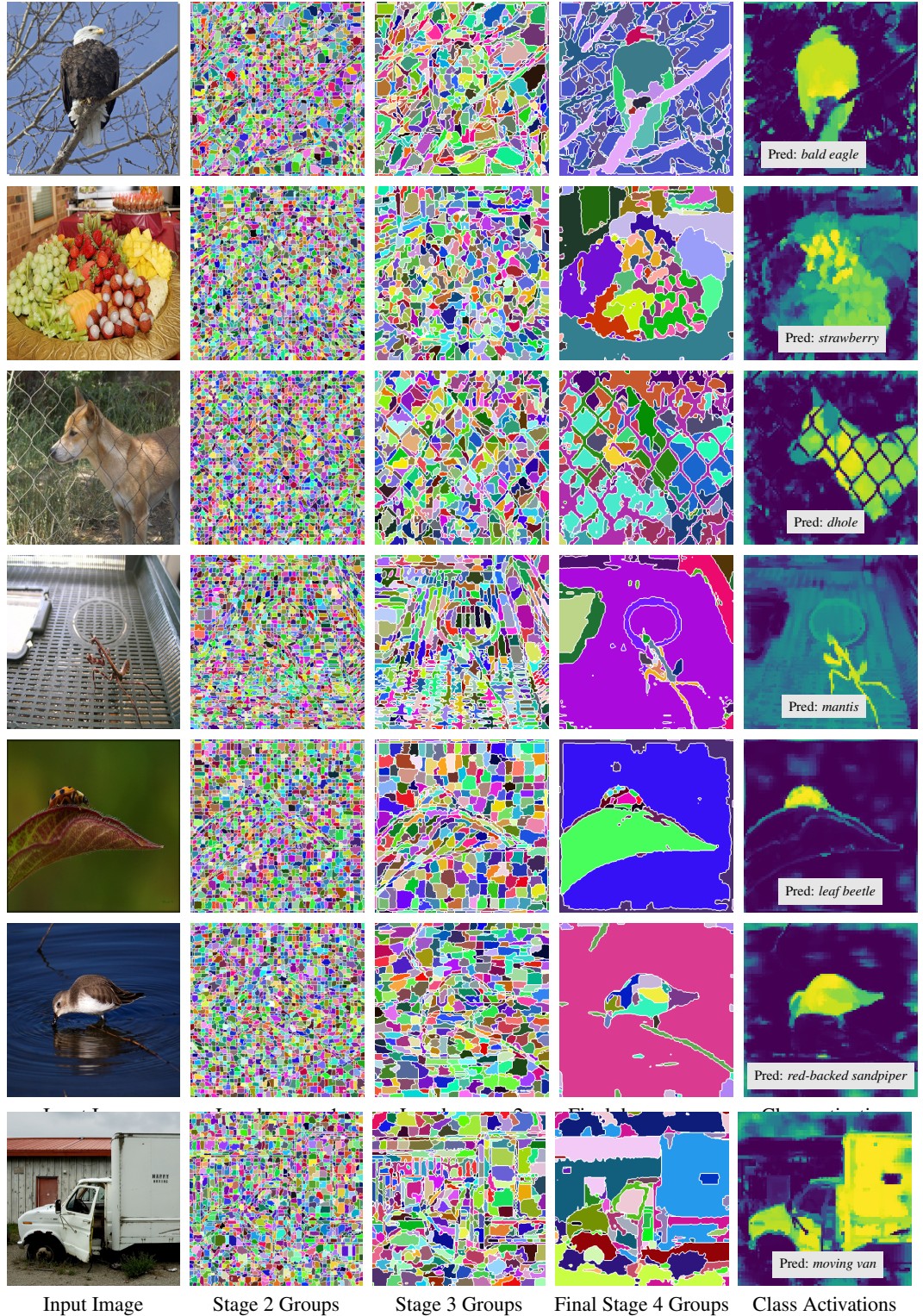

Figure 3: **Segmentation emerges from ImageNet pre-training.** We visualize group decompositions across each backbone stage, along with their upsampled activations over the predicted class. We observe that even in the absence of mask supervision, super-pixel-like structures emerge in earlier layers, and are eventually grouped into semantically coherent regions in dense grouping layers.

| Method | Training data | Postproc. | VOC | Ctx | Obj. | Stuff | City | ADE | Avg. |
|---|---|---|---|---|---|---|---|---|---|
| *CLIP-Pretrained Methods* | | | | | | | | | |
| ViL-Seg [54] | CC12M | - | 37.3 | 18.9 | 18.1 | – | – | – | – |
| SegCLIP [55] | CC3M+COCO | - | 52.6 | 24.7 | 26.5 | – | – | – | – |
| TCL [48] | CC3M+CC12M | PAMR | 55.0 | 30.4 | 31.6 | 22.4 | 24.0 | 17.1 | 30.1 |
| CoDe [56] | CC3M+CC12M | PAMR | 57.7 | 30.5 | 32.3 | 23.9 | 28.9 | 17.7 | 31.8 |
| *Models trained from scratch* | | | | | | | | | |
| GroupViT [9] | CC3M+CC12M+YFCC14M | - | 49.5 | 19.0 | 24.3 | 12.6 | 6.9 | 8.7 | 20.2 |
| ViewCo [57] | CC12M+YFCC14M | - | 52.4 | 23.0 | 23.5 | – | – | – | – |
| CoCu [58] | CC3M+CC12M+YFCC14M | - | 51.4 | 23.6 | 22.7 | 15.2 | 22.1 | 12.3 | 24.6 |
| PGSeg [59] | CC12M+RedCaps12M | - | 53.2 | 23.8 | 28.7 | – | – | – | – |
| SimSeg [44] | CC3M+CC12M | CRF | 57.4 | 26.2 | 29.7 | – | – | – | – |
| **SeNaTra-B (Ours)** | CC3M+CC12M | - | **61.3** | 30.2 | **32.6** | 21.1 | **30.0** | 16.4 | **31.9** |
| **SeNaTra-B (Ours)** | CC3M+CC12M+RedCaps12M | - | **61.4** | **31.2** | **33.2** | 23.2 | **32.1** | 17.4 | **33.1** |

Table 1: **Zero-shot, text-supervised semantic segmentation.** We compare our method to state-of-the-art methods on six datasets, and report average mIoU across datasets where applicable. We bolden top-performers, and underline $2^{nd}$, and indicate postprocessing techniques (CRF [60], PAMR [61]).

image-text pairs, 20× larger than our training set. We observe large improvements (4+) mIoU over all datasets *w.r.t.* methods *not* utilizing CLIP. We note that top-performing methods (TCL [48], CoDe [56], and SimSeg [44]), rely on postprocessing techniques such as PAMR [61] and dense CRFs [60], that increase their performance by $3 - 4$ mIoU , as reported in [48, 44]. In contrast, we obtain strong results due to our network design, without applying any postprocessing. Our approach also surpasses methods leveraging CLIP on most datasets, except ADE20k and COCO-stuff (150 and 133 classes, respectively), where we are second to CoDe. The increased semantic granularity of these datasets benefits from extensive CLIP pre-training. Remarkably, by expanding our training data with just 12M additional image-text pairs from RedCaps12M, we significantly narrowed this gap, showcasing the potential for further scaling.

## 4.2 Training with Mask Supervision

**Overview.** We train SeNaTra with mask supervision on standard semantic [49] and panoptic segmentation [21] datasets. Following common practice, we initialize weights from ImageNet pre-training (Section 4.1.1). Appendix D.3 provides extended results and implementation details.

**Segmentation paradigms.** For each task, we evaluate (i) our minimal *native masks* model that generates masks via backbone-level pixel assignments, and (ii) drop-in backbone replacement in conjunction with a *Mask2Former* (M2F) [6] dedicated head (see Table 2[c]).

*Native segmentation:* We make per-pixel class predictions by feeding our backbone's final group token embeddings through a 2-layer (512 dim.) MLP. We then upsample these (at stride 32) to input resolution using our learned pixel assignments (Section 3.2), and class predictions using cross-entropy loss. For panoptic models, we use an additional 2-layer MLP targeting *objects*. We apply it over the top-100 final group tokens with largest assignment values, representing object candidates. We follow [6] and supervise instance mask and class predictions with a bipartite matching loss [31].

*Ours+Mask2Former:* Our network is versatile and can also be used as a drop-in replacement with networks, such as the widely used M2F, that combine a pixel-decoder using multi-scale deformable attention with a segmentation Transformer decoder. In our version, we replace standard upsampling operations with the assignment matrices obtained through our learned assignments (Section 3.2).

**Baselines.** As *backbone* baselines, we report methods that follow a consolidated design with uniform downsampling, including well-established SwinTransformer [1] and NAT [3], as well as recent bottom-up grouping approaches [11, 12, 10]. We report these in conjunction with dedicated segmentation networks, including: UperNet [40], commonly used for benchmarking vision architectures [1, 3, 25, 62, 63], and widely-used MaskFormer (MF) [13] and Mask2Former (M2F)[6]. We evaluate SeNaTra as both a backbone and to generate *native* masks without dedicated segmentation heads.

### 4.2.1 Semantic Segmentation

**Setting.** We train models to classify pixels into 150 semantic classes on ADE20k dataset [20], and, following common practice, report results on the validation set. We follow similar hyperparameter

| Backbone | Seg. Head | mIoU | #Params | FLOPs |
|---|---|---|---|---|
| *Backbones w/ Uniform Downsampling* | | | | |
| Swin-T [1] | UperNet | 44.5 | 60M | 946G |
| Swin-T [1] | M2F | 47.2 | 47M | - |
| NAT-T [3] | UperNet | 47.1 | 58M | 934G |
| NAT-T* [3] | M2F | 49.1 | 46M | - |
| Swin-B† [1] | M2F | 53.9 | 107M | - |
| Swin-L† [1] | M2F | 56.1 | 215M | - |
| *Backbones w/ Grouping-based Downsampling* | | | | |
| CAST-S [11] | Segmenter | 43.1 | 26M | - |
| TCFm.V1-S [12] | SemFPN | 47.1 | 29M | 370G |
| **SeNaTra-T** | Native | **49.7** | 30M | 113G |
| TCFm.V2-S [12] | M2F | 49.1 | 42M | - |
| ClusterFm.-T [10] | [10] | 49.1 | - | - |
| **SeNaTra-T** | M2F | **51.3** | 47M | - |
| TCFm.V2-B [12] | SemFPN | 50.0 | 66M | 332G |
| **SeNaTra-B** | Native | **51.3** | 95M | 347G |
| TCFm.V2-B [12] | M2F | 53.8 | 80M | - |
| **SeNaTra-B** | M2F | **54.6** | 112M | - |
| **SeNaTra-B†** | M2F | **56.0** | 112M | - |
| **SeNaTra-L†** | M2F | **56.7** | 228M | - |

(a) **Semantic segmentation on ADE20k-val.**

| Backbone | Seg. Head | PQ | #Params |
|---|---|---|---|
| Swin-T [1] | MF | 47.7 | 42M |
| **SeNaTra-T** | Native | **49.2** | 32M |
| Swin-T [1] | M2F | 53.2 | 47M |
| NAT-T* [3] | M2F | 54.3 | 46M |
| ClusterFm.-T [10] | [10] | 54.7 | - |
| **SeNaTra-T** | M2F | **55.0** | 47M |
| Swin-B† [1] | MF | 51.8 | 102M |
| **SeNaTra-B†** | Native | **52.6** | 96M |
| Swin-B† [1] | M2F | 56.4 | 107M |
| **SeNaTra-B†** | M2F | **57.1** | 112M |
| Swin-L† [1] | M2F | 57.8 | 216M |
| **SeNaTra-L†** | M2F | **58.1** | 228M |

(b) **Panoptic segmentation on COCO-val.**

(c) **Segmentation paradigms.**

Table 2: **Downstream semantic and panoptic segmentation after fine-tuning.** **(a)** mIoU on ADE20k. **(b)** PQ on COCO val2017. **(c)** Conceptual visualization of segmentation paradigms. Models marked with † are pre-trained on ImageNet-22K. NAT-T* is our implementation.

configurations as baselines (details in Appendix D.3), except for a reduced number of iterations from $160k$ to $80k$ due to increased convergence speed with our model.

**Discussion.** In Table 2a we observe: (i) our *native* masks yield substantial improvements over both standard and grouping-based backbones using well-established segmentation heads (UperNet [40], Semantic FPN [64], Segmenter [65]), with remarkable compute and parameter-efficiency in our smaller variants. SeNaTra-T achieves $49.7$ mIoU, $+2.6$ *w.r.t.* NAT w/ UperNet ($47.1$ mIoU, NAT-T), with only 12% of its FLOPs and 50% of its parameters. When (ii) using M2F head, our grouping-based representations consistently improve performance across variants: $+1$ mIoU *w.r.t.* M2F + Swin, and $+2.7$ mIoU *w.r.t.* M2F + NAT. Overall, (iii) our backbone adds a modest 5-10% increase in parameters and FLOPs over standard backbones. While combining it with M2F slightly increases computational costs over NAT, this cost is effectively amortized in the native setup where the segmentation head is removed, making the overall approach more parameter- and FLOP-efficient.

#### 4.2.2 Panoptic Segmentation

**Setting.** We train and evaluate models on COCO-panoptic [21], which consists of 80 object (*things*) and 53 background (*stuff*) classes, requiring models to predict semantic classes and instance IDs for *things*. Our models are trained for 50 epochs, using M2F's original hyperparameters for integrated models. For our *native* results, we use the same hyperparameters as in semantic segmentation.

**Discussion.** We observe in Table 2b: (i) our tiny *native* results ($49.2$ PQ) outperform MaskFormer w/Swin-T ($47.7$ PQ) by a sizeable margin, despite fewer parameters ($32M$ *v.s.* $42M$). This trend is consistent across different model sizes, as with Table 2a. (ii) M2F + NAT-T backbone ($54.3$ PQ) outperforms our barebone *native* masks, however, our SeNaTra-T + M2F ($55$ PQ) achieves top performance, and further improves with a larger backbone (SeNaTra-L, $58.1$ PQ). *Overall, our native results surpass consolidated baselines, and our backbone enhances state-of-the-art when paired with dedicated segmentation heads.*

| | S1 | S2 | S3 | ADE20k | ZS-VOC |
|---|---|---|---|---|---|
| Baseline | ✗ | ✗ | ✗ | 41.3 | 40.1 |
| **SeNaTra** | ✗ | ✗ | ✓ | 47.2 | 51.9 |
| | ✗ | ✓ | ✓ | 48.7 | 54.2 |
| | ✓ | ✓ | (local) | 47.3 | 55.8 |
| | ✓ | ✓ | ✓ | **49.7** | **57.3** |

(a) **Impact of grouping at each backbone stage.**

| | ADE20k | ZS-VOC |
|---|---|---|
| **SeNaTra** | **49.7** | **57.3** |
| – Absolute pos. encoding | 48.8 (-0.9) | 56.7 (-0.6) |
| – GRU instead of skip | 44.9 (-4.8) | 55.0 (-2.3) |
| – *nn.Embedding* group init. | 47.2 (-2.5) | 54.1 (-3.2) |
| – Prev. three (Slot Attn. [15]) | 43.6 (-6.1) | 52.3 (-5.0) |

(b) **Low-level design choices in our grouping layer.**

Table 3: **Architecture-level ablations.** We report *native* masks mIoU on ADE20k and Zero-Shot(ZS) mIoU on Pascal VOC. In **(a)**, we evaluate the effect of replacing grouping layers with uniform downsampling at each stage. In **(b)**, we study low-level design decisions inside our grouping layer.

## 4.3 Ablation Studies

**Grouping at different backbone stages.** Table 3a compares our spatial grouping layer to *uniform* downsampling with strided convolution (as in NAT [3], without grouping) over each backbone stage (S1, S2, S3). The *Baseline* underperforms compared to our approach, in both supervised (41.3 mIoU, $-8.4$) and zero-shot (40.1 mIoU, $-17.2$) settings. Instead of learned pixel assignments, this approach relies on bilinear interpolation to predict high-resolution masks from coarse stride 32 feature maps. Moreover, we observe that introducing grouping spatial layers across stages increases performance monotonically. *Local* grouping in the last stage significantly decreases performance in both metrics. Our design enables whole-image masks by leveraging efficient local grouping in early stages.

**Grouping layer design.** Table 3b compares our grouping layer design (Section 3.1) relative to slot attention [15]. Replacing the GRU with skip connections yields an improvement of $+4.8$ mIoU. In practice, we observed that it addressed numerical instabilities during ImageNet pretraining and reduced memory requirements. Similarly, sampling initial embeddings from a learned Gaussian distribution, as in [15], also compromised stability. Using learnable embeddings for initialization, as in [66], still drops performance by $2.5/3.2$ mIoU. Further using relative positional encodings yields an additional 1 mIoU. Altogether, these yield significant improvements of $6.1/5.0$ mIoU on ADE20k and ZS-VOC, respectively, while enhancing training stability and memory footprint.

**Segmentation paradigms.** In Table 4, we ablate: (i) backbone choice (ours with *native* segmentation capabilities vs. baseline [3]), and (ii) two key Mask2Former components: a pixel decoder for multi-scale feature fusion, and a Transformer decoder for producing mask embeddings. In the first two rows, we compare NAT (w/o grouping) with ours, without any additional components. Our baseline fails at this task (PQ 15.9, row 2) and underperforms in semantic segmentation ($-8.4$ mIoU). Adding a pixel decoder (MSDeformAttn from Mask2Former, rows 3&4)

| | Pix Dec. | Tr Dec. | mIoU | PQ |
|---|---|---|---|---|
| **SeNaTra** — *Native* | | | 49.7 | 49.2 |
| — w/o grouping | | | 41.3 | 15.9 |
| **SeNaTra** | ✓ | | 49.7 | 48.8 |
| — w/o grouping | ✓ | | 47.4 | 17.3 |
| **SeNaTra** — M2F | ✓ | ✓ | 51.3 | 55.0 |
| — w/o grouping | ✓ | ✓ | 49.1 | 54.3 |

Table 4: **Segmentation paradigms.** We ablate adding a Pixel Decoder and Transformer Decoder on ADE20k (mIoU) and COCO-Panoptic (PQ).

minimally impacts our approach, but significantly improves NAT baseline ($+6.4$ mIoU). Finally, rows 5&6 show that a segmentation decoder is crucial for NAT to segment instances (54.3 mIoU), and benefits semantic segmentation ($+1.7$ mIoU). Dedicated decoder also benefits our approach in terms of panoptic segmentation (55.0 PQ, $+5.8$ PQ), showing potential for improvement.

## 5 Conclusions

This work introduces a novel architecture particularly suited for segmentation tasks centered around our proposed spatial grouping layer. Our design offers significant methodological advantages over prior art, being fully differentiable with strong inductive bias and scalable to large input resolutions. Through empirical results, we demonstrated the emergence of meaningful segments without explicit mask supervision and a streamlined paradigm for downstream segmentation. Our work shows that segmentation—a fundamental perception task—can be inherently encoded in a model's internal representations rather than delegated to specialized decoder modules, opening new directions in segmentation-centric backbone architectures.

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

# Appendix

**Overview.** We structure the appendix as follows: in Appendix A and Appendix B, we provide a general discussion and broader impact statement, respectively. In Appendix C we show visualizations of our model's learned hierarchical decompositions and final segmentation masks for image-text supervision. In Appendix D we provide extensive implementation details and additional experimental results. Lastly, in Appendix E we discuss low-level details on our spatial grouping layer's implementation and its runtime and compute considerations.

## A   Discussion

SeNaTra introduces a new family of backbone architectures enabling native segmentation through spatial grouping layers. As demonstrated in our experiments, our approach outperforms strong baselines and previous grouping-based works both in the purely native setting (including zero-shot), as well as with additional segmentation heads. Despite its promising results, SeNaTra has limitations. While our model scales approximately linearly with respect to input resolution (see Appendix E.2), grouping layers introduce computational overhead compared to their de-facto counterpart, strided convolutions, given the lightweight nature of the latter. This added complexity is largely amortized when leveraging native segmentation capabilities but remains a limitation when integrating our model with external heads. As explained in Appendix E.1, we provide an efficient CUDA-based implementation, however, there remains room for improvements both in terms of low-level CUDA optimizations and general module design. Another consideration is that while our native results perform favorably in both semantic and panoptic segmentation, our model yields larger gains in semantic segmentation. A plausible explanation lies in biases acquired during ImageNet pre-training. During this stage, our model is not incentivized to separate different instances of the same class in grouping layers, but rather to focus on overall semantics. Therefore, the adaptation needed for a pre-trained model to transfer knowledge through grouping layers for semantic segmentation is likely smaller than that required for panoptic segmentation, where instance separation is required. Throughout our experiments, we focused on using off-the-shelf pre-training recipes to highlight advantages induced solely by our architectural design. However, we believe there are multiple exciting opportunities for future work to address this observation and design object-oriented pre-training schemes, such as work focusing on visual grounding [67].

## B   Broader Impact Statement

Our work proposes a new vision backbone architecture with applications mainly in the field of segmentation. Given the broad scope and potential applications of this task, and computer vision systems overall, our model inherits both opportunities and challenges common to this field. Like any general-purpose data-driven model, it may exhibit biases present in training data and could potentially be adapted for concerning applications. However, some of our empirical results demonstrate improved parameter and compute efficiency compared to existing solutions, as well as increased data efficiency. These properties could enable positive impact in resource-constrained settings where access to large datasets or computational resources is limited, such as applications in life sciences.

## C   Zero-Shot Qualitative Results

In Figure 4, we show both per-stage groups as well as final predicted semantic masks for SeNaTra-B pretrained on image-text pairs, *i.e.*, CC3M and CC12M datasets. The results are obtained on validation images of the PASCAL VOC dataset [49]. As with class-supervised models, we observe a hierarchy of boundary-preserving groups across stages. We notice that our model's final groups have a larger tendency towards oversegmenting objects and regions. This can be explained due to the richer and *denser* semantic content present in text embeddings, which may benefit from a more granular visual representation. Remarkably, by querying final tokens with text embeddings we obtain high-quality semantic masks (column 5), suggesting that our model's pixel partitions carry semantic awareness.

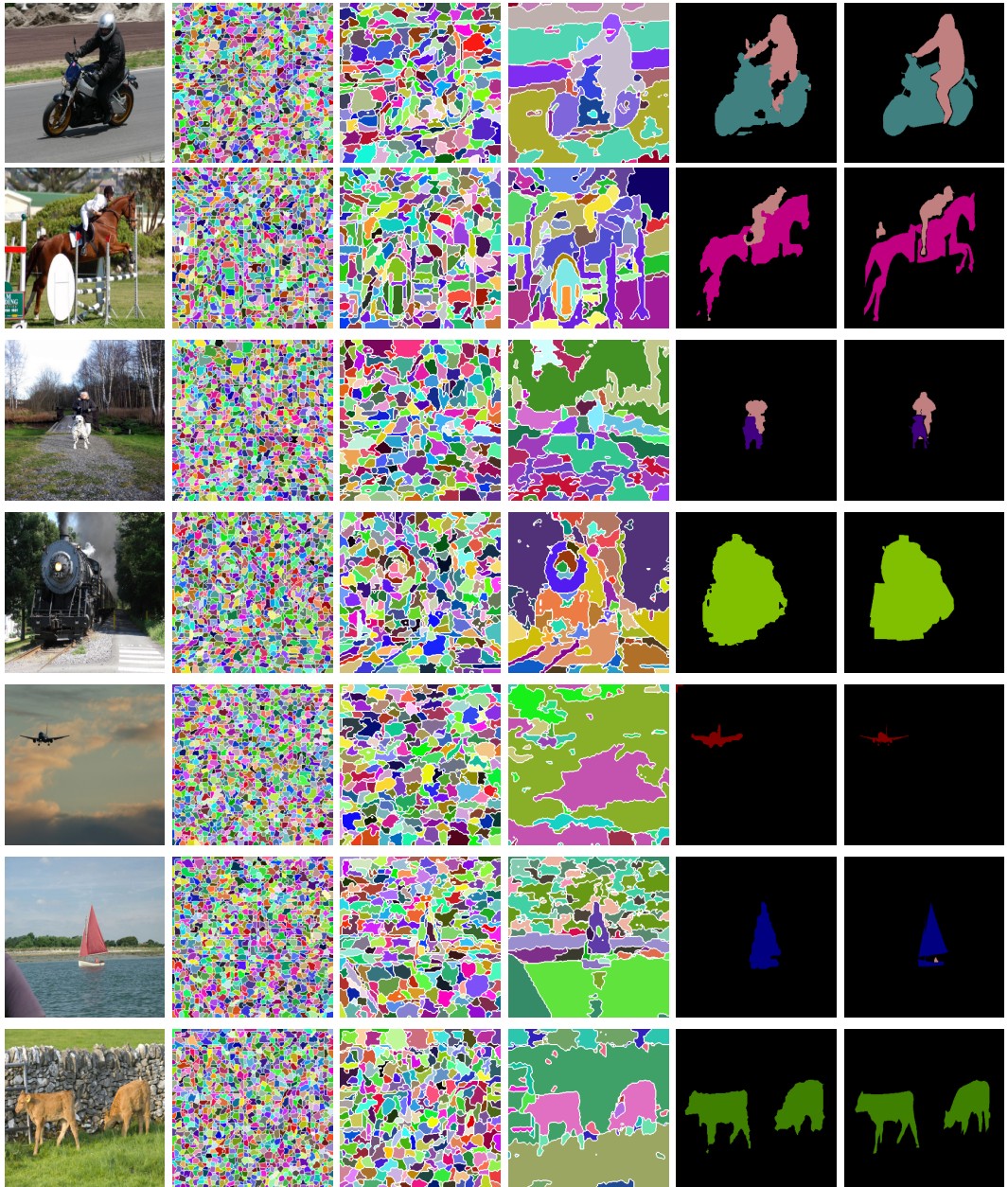

Input Image     Stage 2 Groups     Stage 3 Groups   Final Stage 4 Groups   Predicted Masks   Ground Truth Masks

Figure 4: **Qualitative zero-shot segmentation learned from image-text contrastive pre-training.** We visualize hierarchical final decompositions along with their predicted semantic masks, obtained in a zero-shot setting on Pascal VOC validation images [49], and corresponding ground truth masks. Note that these models did not receive any form of mask supervision during training, and were trained with a standard contrastive objective on image-text pairs. Final masks are obtained without any form of heuristic postprocessing.

## D   Additional Results and Implementation Details

**Overview.** In the following, we provide additional results, and implementation details for each of the experimental setups described in the main paper. *Note that our code and pre-trained models will be made publicly available*.

**Model variants.** As explained in Section 4.1, we present results with three different model variants, each corresponding to an increased parameter count. In Table 5, we specify the configuration of each variant, including (i) number of transformer encoder layers, *i.e.*, blocks consisting of self-attention

| | # layers | dim | MLP ratio | # params |
|---|---|---|---|---|
| SeNaTra-T | 3, 4, 18, 5 | 512 | 3 | 29M |
| SeNaTra-B | 3, 4, 18, 5 | 1024 | 2 | 94M |
| SeNaTra-L | 3, 4, 18, 5 | 1536 | 2 | 211M |

Table 5: **Model variants.** We summarize the model configuration of each of our backbone variants: number of transformer encoder layers used at each stage (*# layers*), output token dimension (*dim*), and MLP hidden dimension ratio in transformer encoder layers (*MLP ratio*).

followed by an MLP with residual connections, (ii) final embedding dimension, and (iii) MLP ratio, *i.e.*, hidden dimension of the aforementioned MLPs in transformer encoder layers. The overall configuration used for each model variant is borrowed from [3], with the exception of *large*, which was not presented in its original work. We choose the design of [3], due to its strong baseline performance among existing networks in both classification and dense prediction.

**Rotary Positional Encodings.** In the original work of [3], the self-attention layers used at each backbone stage were implemented with relative positional encodings [68]. Recently, Rotary Positional encodings (RoPe) [69, 70] have gained popularity, due to their ability to encode pairwise positional information in a principled way and without the need for explicit access to the self-attention matrix. This can be largely beneficial in terms of runtime and memory consumption, as it allows the use of fused implementations of self-attention [71, 72], resulting in significant runtime improvements. For the sake of efficiency, we replace the relative position biases in self-attention layers originally used by [3] with RoPe, and leverage the recently proposed fused kernels from [72]. Empirically, we observe a negligible decrease of downstream classification and segmentation performance, and a significant increase in speed. Particularly on newer hardware, *e.g.*, A100s, this change results in an approx 30% decrease in runtime, helping offset the overhead introduced by our spatial grouping layers.

### D.1 ImageNet Classification

**ImageNet-1k pre-training: implementation details.** We train our models from scratch for 300 epochs at resolution $224 \times 224$, following all training hyperparameteres, including optimizer, learning rate scheduler, and augmentation setting of [1]. However, we disable MixUp augmentation as it degrades our results. The drop is likely caused by the ambiguity that alpha composite images introduce in our grouping layer. Unlike [3], we do not train for additional *cooldown* epochs. Following [1, 3], we use stochastic depth for regularization [74], with default survival probabilities of 0.3 and 0.5 for our *tiny* and *base* variants, respectively. Trainings take approximately 36 hours on 8 A100 GPUs.

**ImageNet-22k pre-training: implementation details.** Following [1, 25], we pre-train our larger model variants for 90 epochs on the larger-scale ImageNet-22k dataset, which consists of approximately 16M images labeled over $22k$ classes. In this setup, we reduce stochastic depth

| method | image size | #params. | FLOPs | top-1 acc. |
|---|---|---|---|---|
| *ImageNet-1k trained models* | | | | |
| ViT-B/16 [24] | 384² | 86M | 55.4G | 77.9 |
| DeiT-S [73] | 224² | 22M | 4.6G | 79.8 |
| DeiT-B [73] | 224² | 86M | 17.5G | 81.8 |
| ClusterFormer-T [10] | 224² | 28M | - | 81.3 |
| Swin-T [1] | 224² | 29M | 4.5G | 81.3 |
| ConvNeXt-T [25] | 224² | 28M | 4.5G | 81.3 |
| TCFormerV2-S [25] | 224² | 26M | 4.5G | 82.4 |
| NAT-T [3] | 224² | 28M | 4.5G | **83.3** |
| **SeNaTra -T** (Ours) | 224² | 29M | 4.9G | 83.1 |
| Swin-B [1] | 224² | 88M | 15.4G | 83.5 |
| ConvNeXt-B [25] | 224² | 89M | 15.4G | 83.8 |
| NAT-B [3] | 224² | 90M | 13.7G | **84.3** |
| **SeNaTra -B** (Ours) | 224² | 90M | 14.9G | 84.0 |
| *ImageNet-22k pre-trained models* | | | | |
| ViT-B/16 [24] | 384² | 86M | 55.4G | 84.0 |
| Swin-B [1] | 384² | 88M | 47.0G | 86.4 |
| ConvNeXt-B [2] | 384² | 89M | 45.1G | 86.8 |
| **SeNaTra -B** (Ours) | 384² | 94M | 47.9G | **86.9** |
| ViT-L/16 [24] | 384² | 307M | 190.7G | 85.2 |
| Swin-L [1] | 384² | 197M | 103.9G | 87.3 |
| ConvNeXt-L [2] | 384² | 198M | 101.0G | **87.5** |
| **SeNaTra -L** (Ours) | 384² | 211M | 107.3G | 87.3 |

Table 6: **Image classification on ImageNet-1k and -22k.** We compare various standard and grouping-based backbones both trained from scratch on 1k and pre-trained on 22k.

probabilities to 0.2, following [1]. We follow prior art [1, 25] and fine-tune these models for 30 additional epochs on the ImageNet-1k dataset at $384 \times 384$ resolution and report top-1 accuracy

on ImageNet-1k-val. As in the previous setup, we follow the training recipe and parameters of [1], both for pre-training and fine-tuning, with the exception of MixUp augmentation. Pre-training runs are conducted on 16 A100 GPUs for approximately a week, and fine-tuning on ImageNet-1k takes approximately 6 hours on 8 A100s.

**Discussion.** In Table 6, we provide quantitative results of our model against state-of-the-art methods following a comparable supervised setup. As noted in the main text, our approach performs on par with state-of-the-art, while enabling the emergence of strong pixel-level localization properties, as it can be observed qualitatively in Figure 3.

### D.2   Image-Text Pretraining

**Pre-training: implementation details.** As explained in the main paper, we adhere to the training setup of [44]. We note, however, that [44] open-sourced its evaluation code, but did not provide scripts nor instructions to train its model. Our attempts at reproducing their reported results lag $1 - 3$ mIoU behind the reported numbers. Following the hyperparameters specified in their original paper, we train for 20 epochs with initial learning rate $3 \times 10^{-4}$ and a use a cosine decay scheduler leading to a minimum learning rate of $3 \times 10^{-5}$. We apply linear learning rate warmup during the first $3k$ iterations, and train for a total of 20 epochs with a batch size of $4096$ (approx. $68k$ iterations). As with ImageNet models, we use stochastic depth for regularization [74], with survival probabilities set to $0.2$ and $0.3$ for our *tiny* and *base* model, respectively. For our configuration using additional data from RedCaps12M, we keep all hyperparameters identical, and train for 20 epochs on the union of all datasets (CC3M, CC12M, and RedCaps12M), leading to a total of $126k$ iterations. Trainings are conducted on 16 A100 GPUs, lasting approximately two days for default configurations, around four days when using the RedCaps12M dataset.

**Backbone-level comparison.** As explained, we follow the experimental setup described [44] and replace its backbone (ViT, [24]) with our proposed SeNaTra backbone. We highlight the impact of this change in Table 7. We note that SimSeg [44] produces coarse patch-class activations and relies on postprocessing with Conditional Random Fields [60] to obtain pixel-precise masks. Our method does not require such heuristic postprocessing and instead utilizes our upsampling operations (Section 3.2) to produce pixel-level output.

| Backbone | CRF | VOC | Context | COCO-Obj. | Avg. |
|---|---|---|---|---|---|
| ViT-S | ✗ | 53.8 | 23.5 | 25.7 | 34.3 |
|  | ✓ | 56.4 | 25.8 | 27.2 | 36.5 |
| **SeNaTra -T** | ✗ | **57.3** | **27.3** | **29.7** | **38.1** |
| ViT-B | ✗ | 53.1 | 23.3 | 27.4 | 34.6 |
|  | ✓ | 57.4 | 26.2 | 29.7 | 37.8 |
| **SeNaTra -B** | ✗ | **61.3** | **30.2** | **32.6** | **41.4** |

Table 7: **Backbone comparison for text-supervised zero-shot segmentation.** Our approach significantly outperforms SimSeg [44] trained with a ViT backbone, without relying on CRF post-processing.

For *base*, our approach yields a $3.5$ avg. mIoU increase, which further increases to $6.7$ mIoU when comparing postprocessing-free outputs.

**Zero-shot Segmentation Inference.** As explained in the main text, we quantitatively evaluate the performance of our image-text pre-trained models on *zero-shot semantic segmentation*. Our inference details are the following: given a dataset with $C$ target classes, we obtain $C$ corresponding text embeddings by feeding template prompts as *"An image of* {CLASS }*"* through our text encoder (we use the original templates of [44]). We then compute the dot-product similarity between each of our $N^{\text{out}}$ final projected image tokens and target class embedding. By repeating this process over all $C$ text embeddings, we obtain an $N^{\text{out}} \times C$ unnormalized similarity map, which we upsample to input resolution, *i.e.*, patch-level, with the transition matrices described in Section 3.2. By applying an argmax over classes, we obtain a final class prediction for each input patch. For datasets with an additional *background* class (Pascal VOC [49], Pascal Context [50] and COCO-obj [51]), we need to set a threshold for mask values. To do so, we use the original method of [44], consisting of computing the image-level-text embedding similarity of the top-k classes in the dataset, and computing the mean value with an additional standard deviation. For the remaining datasets, we just apply a pixel-wise argmax over all classes. Lastly, we obtain a small performance boost by computing overall class similarities as the average of image-level similarities (after pooling), and the maximum spatial similarity over final tokens, *i.e.*, max similarity over our final groups.

## D.3 Native Segmentation Models with Mask Supervision

**Semantic Segmentation: implementation details.** As explained in the main paper, our *native segmentation* for semantic segmentation obtains masks by feeding final token embeddings through a $2-layer$ MLP, followed by upsampling the results to input resolution with our learned upsampler. Our model is implemented with `mmsegmentation`. For simplicity, we follow the default configuration of the original FCN [27], which further concatenates final token embeddings after feeding them through the MLPs, akin to skip connections. We have not explored this design thoroughly and better alternatives are likely possible. Further following the original configuration of [27], during training we add an auxiliary loss at our penultimate stage, consisting of an additional MLP that's used analogously to the one one in our final layer. This auxiliary MLP is ignored at test-time. Models are trained with the same training hyperparameters used by Swin Transformer used in conjunction with UperNet, with the only difference being an increased weight decay of $0.05$, and a shorter schedule of $80k$ iterations, instead of the original $160k$. Trainings are conducted on $8$ A100 GPUs for approximately 6 hours.

**Panoptic Segmentation: implementation details.** As explained in the main paper, we use two main MLPs for panoptic segmentation targeting *things* and *classes* separately. The logic behind this division is to avoid penalizing oversegmentation errors for background regions. Therefore we use the MLP targetting *stuff* regions as in our semantic segmentation model and produce class predictions over each input token individually. The MLP targeting *things* is only applied over the top $100$ final tokens with largest assignment values in the last stage, representing potential object candidates. This MLP classifies tokens into either a *things* class label or *no object*. While it is possible to obtain a mask for each object candidate by directly using its corresponding input-level assignment, we find it beneficial to refine instance mask predictions by computing the dot-product between object candidates' final embeddings and our penultimate stage output embeddings, effectively recomputing the assignment matrix in our last grouping layer. In addition, before computing the dot-product, we linearly project final token embeddings and upsample them to the previous stage resolution, aiming to provide global context to Stage 3 features before re-computing the assignment. This procedure enables correcting common over-segmentation errors and introduces no significant overhead. We supervise the resulting masks and class predictions with the same bipartite matching loss as [6]. Lastly, during training, we find it beneficial to apply both MLPs and their corresponding losses to the intermediate outputs of our last stage (5 in total), akin to the intermediate losses used by [31, 13, 6]. At test-time, intermediate predictions are not used. Lastly, models are trained on $8$ A100s following the same configuration from our semantic models, but for a longer schedule of $50$ epochs—as in Mask2Former [6]–and taking around $2.5$ days.

# E Efficiency and Performance Considerations

## E.1 Efficient Sparse Implementation

**Naive implementation.** In Algorithm 1, for clarity when describing our grouping layer, we depict a naive implementation with sparsity constraints. Formally, $M_{\texttt{loc}} \in \{0, -\infty\}^{N \times N^{\text{down}}}$ is defined as 0 for input-output *edges* that are enabled (see Figure 2), and $-\infty$, *i.e.*, a large negative constant, for the rest. By being added to $A$ in L4, right before applying `softmax` in L5, entries set to $-\infty$ effectively become 0. In this setup, dense grouping naturally corresponds to $M_{\texttt{loc}}$ containing zeros in all of its entries. This formulation accomodates both local and dense grouping, but is not the one we utilitze in practice.

**Optimized implementation.** The problem with the naive implementation described in Algorithm 1 is that given $N^{\text{in}}$ input tokens and $N^{\text{out}}$ output, tokens, it requires storing a dense $N^{\text{in}} \times N^{\text{out}}$ output matrix, which becomes impractical for input sets with large cardinality as the ones we process in the early layers of our backbone. To exploit sparsity and avoid computing non-zero entries in the cross-attention matrix, we leverage the sliding window attention CUDA kernels introduced in the `natten` [3] library. In Algorithm 2, we outline the high-level implementation of the sparse variant of the cross-attention and re-normalization operations described in Algorithm 1 (L3-8, excluding the use of LN) using *PyTorch*. Abusing notation, we denote $q = q(X^{\text{out}})$, $k = k(X^{\text{in}})$, $v = v(X^{\text{in}})$, corresponding to linearly projected output and input token embeddings. The main idea of the the implementation is to repurpose the `natten` primitives `na2d_qk`, corresponding to query-key cross-attention multiplication, and `na2d_av`, corresponding to computing a weighted sum of values from the query-key matrix.

To do so, given our input feature tensor $X^{\text{in}}$, with spatial dimensions $H \times W$, and an initial set of of output downsampled tokens $X^{\text{out}}$ with spatial dimensions $(H/2) \times (W/2)$, we first Unfold $X^{\text{in}}$ into $2 \times 2$ patches, resulting in a $4 \times (H/2) \times (W/2)$ tensor. Now by *expanding*, *i.e.*, creating view with 'multiple copies', our target downsampled tokens $X^{\text{out}}$ into $4 \times (H/2) \times (W/2)$, and linearly projecting them as described, we obtain a tensor representation in which applying sliding window *cross-attention* yields our expected grouping operation over local windows. In Algorithm 2, we further refer to gather operations based on q_idx (L6). At a high-level, q_idx is an index tensor that enables rein-

---

**Algorithm 2** Efficient implementation of sparse Spatial Grouping Layer cross-attention operation

**Input:** $k, q, v,$ q_idx $, B, \tau, \epsilon$
1: // Compute sparse cross-attention
2: attn $\leftarrow$ na2d_qk$(k, q, B, 3)$
3: // Softmax over "groups"
4: attn $\leftarrow$ softmax$(\text{attn} \times \tau, \dim = -1) + \epsilon$
5: // Reindex, reshape, renormalize over inputs
6: attn_q $\leftarrow$ gather$(\text{attn.flatten}(2, 3), \text{q\_idx})$
7: denom $\leftarrow$ sum$(\text{attn\_q}, \dim s = (1, 3))$
8: attn_q $\leftarrow$ attn_q/denom
9: attn_q $\leftarrow$ reshape$(\text{attn\_q}, \text{attn.shape})$
10: // Aggregate updates
11: updates $\leftarrow$ na2d_av$(\text{attn\_q}, v, 3)$
12: updates $\leftarrow$ sum$(\text{updates}, \dim = 1)$
13: **return** updates, attn, attn_q

---

dexing the cross-attention matrix produced in L2 to enable computing the weighted mean from groups over inputs efficiently (corresponding to Algorithm 1, L6-8). The resulting sparse matrix attn_q can then be used as needed in na2d_av. These same ideas enable us to efficiently upsample and downsample feature maps with our resulting assignment matrices by leveraging the aforementioned primitives.

While it is possible to further optimize these operations by defining *fused* kernels that merge the computation of cross-attention, renormalization, and reindexing without storing intermediate tensors, our proposed implementation already provides a large improvement over naive implementations, or pure PyTorch-based operations based on *Unfold*, as we show in Table 9. Overall, while our current implementation not fully optimal it enables the practical usage of our grouping layer within modern backbones on large resolution feature maps without exploding memory requirements. As already mentioned, *we will release our code and models*.

### E.2 Runtime and Memory Analysis

| Model | Seg. Head | FPS | Time (ms) | Mem. (GB) | mIoU |
|---|---|---|---|---|---|
| NAT-B [3] | UperNet [40] | 37.4 | 26.7 | 0.9 | 48.5 |
| SeNaTra-B | Native | 43.6 | 22.9 | 0.6 | **51.3** |

Table 8: **End-to-end model performance and resource usage.** We compare our native masks against the NAT baseline (with a UPerNet [40] decoder) in terms of throughput, latency, GPU memory, and final downstream mIoU on ADE20k.

**Setup.** We evaluate the computational efficiency of our method on an NVIDIA A100 GPU with 40GB of VRAM using batch size one and full FP32 precision for our base model variant across multiple input resolutions in Table 8, and at standard $512 \times 512$ for our Table 9.

**Discussion.** In Table 8 we compare three implementation approaches: conventional uniform downsampling (None, equivalent to our NAT [3] baseline), a naive, pure PyTorch implementation of our grouping layer (Naive), and our CUDA-optimized implementation, leveraging natten (CUDA). Our CUDA-optimized spatial grouping enables practical deployment at high resolutions—where the naive implementation becomes prohibitively slow beyond

| Resolution | Impl. | FPS | Time (ms) | Mem. (GB) |
|---|---|---|---|---|
| $256^2$ | None | 72.9 | 13.7 | 0.4 |
| | Naive | 49.7 | 20.1 | 0.6 |
| | CUDA | 51.0 | 19.6 | 0.4 |
| $512^2$ | None | 57.8 | 17.3 | 0.4 |
| | Naive | 17.9 | 55.8 | 3.2 |
| | CUDA | 44.7 | 22.4 | 0.5 |
| $768^2$ | None | 36.7 | 27.2 | 0.5 |
| | Naive | 6.8 | 146.8 | 14.8 |
| | CUDA | 29.0 | 34.5 | 0.6 |
| $1024^2$ | None | 23.3 | 43.0 | 0.7 |
| | Naive | – | – | OOM |
| | CUDA | 18.2 | 54.9 | 0.7 |

Table 9: **Backbone-level throughput and resource usage.** We report FPS, per-image latency, and peak GPU memory for different input resolutions and grouping implementations. "OOM" indicates out-of-memory.

$512 \times 512$ and runs out of memory on a 40GB GPU at $1024 \times 1024$ resolution. Critically, our local grouping design ensures that both memory consumption and runtime scale approx. linearly with input resolution, making our approach practical for real-world applications. While our method introduces a latency overhead ranging approx. $20 - 40\%$ compared to uniform downsampling baselines when used solely as a feature extractor (Table 9), the overall cost of our grouping layers is amortized when considering end-to-end segmentation performance, as shown in Table 8: our native segmentation capability eliminates the need for heavy decoder heads, ultimately reducing overall latency while simultaneously improving segmentation quality (mIoU). Moreover, the relative overhead of grouping layers is reduced at higher resolutions, due to the scalability of our approach, and the relatively smaller compute being dedicated to downsampling vs. the rest of the backbone. In future work, additional optimizations over our implementation and design could further bridge this gap.

