# OpenReview forum: "Native Segmentation Vision Transformers"
_NeurIPS.cc/2025/Conference — NeurIPS 2025 poster_

### Official Review · Reviewer_Z1HJ · 2025-06-11

**Clarity:** 3
**Significance:** 3
**Originality:** 3
**Rating:** 4
**Confidence:** 3

**Summary:**

This paper proposes a novel segmentation architecture Native Segmentation Vision Transformer (SeNaTra), which replaces traditional uniform downsampling in vision backbones with content-aware spatial grouping layers. The spatial grouping layer creates hierarchical segmentation that emerges naturally during feature extraction, eliminating the strict need for dedicated segmentation heads.

**Questions:**

See weakness

**Ethical Concerns:**

["NO or VERY MINOR ethics concerns only"]

**Final Justification:**

The response of author have been addressed my all concerns

**Quality:**

3

**Strengths And Weaknesses:**

**Strengths**
1. This paper is well-organized and easy to follow.
2. The paper propose a novel architecture that replaces grid-based uniform downsampling with semantic-aware spatial grouping, intrinsically encoding segmentation capabilities within the backbone's hierarchical feature extraction without requiring dedicated segmentation heads.
3. Significant outperformance on zero-shot segmentation tasks, even against models trained on much larger datasets.

**Weaknesses**
1. While the "native" model is effective, the paper's best results still rely on a powerful segmentation head (Mask2Former). This weakens the argument for decoder-independence and demonstrates that such heads remain essential for performance.
2. The paper would be strengthened by a discussion and empirical comparison with ToMe [1].
3. How sensitive is the model to the number of grouping iterations L, a key hyperparameter that was fixed at 3? An ablation study analyzing the performance vs. efficiency trade-off for varying L would be a valuable addition.


[1] Token Merging: Your ViT But Faster, ICLR 2023

---

> ### Author Rebuttal · Authors · 2025-07-31
>
> We thank the reviewer for their constructive feedback and overall positive assessment of our approach, highlighting its architectural novelty and presentation clarity. We address the reviewer's comments below.
>
> **The reviewer points out that a dedicated segmentation head is still needed to achieve state-of-the-art performance for downstream segmentation results.**
>
> The reviewer correctly points out that our best performance on downstream segmentation tasks is still achieved by combining our backbone with Mask2Former (M2F). However, we would like to highlight the following points:
> - In  Table 2(a), we show that for Semantic Segmentation, the gap is already relatively small. For instance, Ours-T + native only lacks 1.6 mIoU behind Ours-T + M2F, despite a ~36% reduction in trainable parameters.
> - While in Table 2(b) we observe larger gaps between native and M2F-based results for our backbone for Panoptic Segmentation, we would like to highlight that M2F is a highly optimized model that has benefited from significant architectural contributions over its predecessor, MaskFormer. In contrast, in our work, we show a first iteration of our novel lightweight native paradigm, with streamlined design choices. Despite its minimalistic nature, it already outperforms a strong baseline such as MaskFormer, which we believe is a significant milestone.
>
> Overall, we aim for our results to show the potential of our native paradigm, and believe that with community adoption and similar future contributions, such as those applied over the original MaskFormer, native architectures will be able to bridge this gap.
>
> **The reviewer requests a discussion of ToMe**
>
> We thank the reviewer for pointing out the "Token Merging: Your ViT But Faster" (ToMe) paper. We will include a dedicated discussion in our revised paper's related work section to clarify its relationship to our approach. We highlight the most important differences below:
> - Distinct Focus: *Segmentation vs. Classification Throughput*. ToMe primarily aims to increase throughput with minimal accuracy degradation for existing ViTs by heuristically merging tokens for image/video classification tasks, specifically showing results on ImageNet, K400, and AudioSet-2M. It does not report results for downstream segmentation, which is the key focus of our work. In contrast, our SeNaTra introduces a novel architectural paradigm based on differentiable, content-aware spatial grouping directly within hierarchical backbones, comprehensively evaluated on multiple downstream segmentation tasks (semantic and panoptic).
> - Architectural Integration for Native Segmentation. ToMe is a "drop-in" method that can be applied to existing ViTs without retraining, merging tokens in each block based on similarity. While efficient for throughput, it's an optimization applied to pre-existing architectures. Our work fundamentally replaces the uniform downsampling operation in hierarchical backbones with learned grouping layers. This architectural innovation enables "native segmentation," meaning masks emerge directly from the backbone, offering a more intrinsically segmentation-aware representation, with benefits across zero-shot and downstream segmentation tasks.
> - Comparison Challenges. Directly comparing ImageNet classification results is difficult because ToMe primarily reports performance on pre-trained ViTs using mostly self-supervised recipes (including MAE fine-tuning), whereas our ImageNet pre-training for segmentation is standard classification-based. This prevents a truly "apple-to-apple" comparison in that domain. Moreover, as already mentioned, ToMe's focus is not on accuracy improvements but rather on throughput improvement, which is beyond our scope.
>
> Overall, while both approaches explore token reduction in vision transformers, their scope and primary objectives are orthogonal. ToMe optimizes throughput for classification in existing models while aiming for minimal accuracy drops, while our work focuses on a new architectural paradigm for native, data-efficient segmentation, offering foundational benefits for the development of future segmentation-centric models. We will include this discussion in the revised paper.
>
> **The reviewer asks about the impact of the number of grouping iterations.**
>
> The reviewer correctly points out that the number of iterations for grouping was fixed at 3 during our experiments, and ablating this hyperparameter could be a valuable addition to our work. Below, we provide an experiment where we analyze the role of this parameter for both downstream Semantic Segmentation with fine-tuning on ADE20k, and zero-shot segmentation from text-supervision on PASCAL VOC, as in Table 3, and retrain models with a varying number of iterations. To study efficiency, we additionally report latency (in ms) during inference for a SeNaTra-B model using batch size 1 and on 512x512 images, measured on an NVIDIA A100 GPU (using fp32 precision). Results are summarized below:
>
> | \# iterations | ADE20k (w/ fine-tuning) | Pascal VOC (zero-shot) | Latency (ms) |
> |--------------|-------------------------|------------------------|--------------|
> | 1            | 48.4                    | 55.5                   | 19.1         |
> | 2            | 49.5                    | 56.9                   | 20.8         |
> | 3            | 49.7                    | 57.3                   | 22.4         |
>
>
> Overall, we observe that our models obtain the highest segmentation accuracy when performing three grouping iterations. This indeed introduces a moderate overhead compared to fewer iterations; however, this parameter can always be adjusted for downstream applications that require a different tradeoff between segmentation accuracy and latency. We will expand on this this analysis in our final manuscript.

---

> > ### Comment · Reviewer_Z1HJ · 2025-08-01
> >
> > Thank you for the response. My concerns have been addressed.

---

### Official Review · Reviewer_RCbm · 2025-06-18

**Clarity:** 4
**Significance:** 4
**Originality:** 4
**Rating:** 5
**Confidence:** 5

**Summary:**

The paper challenges the current standard for spatial downsampling in vision backbones, replacing uniform downsampling with a learnable image-adaptive downsampling based on differentiable grouping. Thanks to this novel spatial grouping layer, super-pixel-like naturally arises in feature maps, enabling zero-shot segmentation tasks without additional task-specific heads or mask supervision. Being a general backbone, the proposed method can also be improved by leveraging mask supervision and segmentation heads, while not being strictly required.

**Questions:**

Coming back to the previously mentioned weakness, for example, having the backbone native segmentation capabilities, I would have liked to see if lighter segmentation heads (e.g. "You Only Segment Once: Towards Real-Time Panoptic Segmentation" or "PEM: Prototype-based Efficient MaskFormer for Image Segmentation".) could benefit more from this paradigm wrt the heavy M2F head. Since the backbone natively supports segmentation, I suppose the heads would need to do much less heavy-lifting.

I wonder how this approach, leading to native segmentation arised thanks to architectural design, could be leveraged together with vision foundation models (e.g. DINO) where similar properties wrt dense predictions arise through training, considering that these models are usually ViT-based and non-hierarchical.

**Ethical Concerns:**

["NO or VERY MINOR ethics concerns only"]

**Final Justification:**

Amazing new architecture with lot of possible follow up works. It can be a groundbreaking innovation in segmentation.

**Limitations:**

The only limitation, as also stated in the paper, is the additional overhead introduced by the grouping layer. Indeed, this is likely a minor limitation, since this overhead is negligible wrt the overhead introduced by task specific modules, not strictly needed anymore for this model.

**Paper Formatting Concerns:**

No issues, there may only be a typo in the labels at the bottom of Figure 5 in the supplementary material, where the labels overlap.

**Quality:**

4

**Strengths And Weaknesses:**

Strengths:
- the paper is well written and easy to follow;
- it challenges the current standard in vision, proposing a novel spatial downsampling technique;
- segmentation naturally arises from the architecture, without mask-level supervision;
- solid experimental section and ablation studies;
- this new method can either improve existing segmentation pipelines or pave the way for new paradigms not using task-specific heads.

Weakness:
- the only minor weakness I can find is the limited set of competitors included in table 2, many others backbones and segmentation heads could be included.

---

> ### Author Rebuttal · Authors · 2025-07-31
>
> We appreciate the reviewer’s thorough evaluation and positive feedback on our paper, methodology, and “comprehensive evaluation”. We address the reviewer’s questions below:
>
> **The reviewer asks for a comparison with additional lightweight segmentation heads.**
>
> The reviewer notes that additional backbones and segmentation heads could strengthen Table 2 further. In its current version, we focused on representative models, including widely used transformer-based backbones (e.g., Swin) and established heads (e.g., Mask2Former). Below, we provide results on COCO-Panoptic with the additional head proposed: YOSO. For our implementation, we replace transposed convolution layers in the pixel decoder with our learned upsampling operators. Results are summarized in the table below, where we include our native and Mask2Former-based results as references.
>
> | Backbone | Head        | PQ   |
> |----------|-------------|------|
> | Ours-T   | Native      | 49.2 |
> | Ours-T   | YOSO        | 52.1 |
> | Ours-T   | Mask2Former | 55.0 |
>
>
> We observe that YOSO partially bridges the gap between our native setup and Mask2Former. These results align with the reviewer’s intuition, showing that with our backbone, this lightweight head can perform favorably, providing middle-ground between the minimal native paradigm and the heavier Mask2Former head.
> We commit to including additional heads (including PEM, as suggested by the reviewer) in Table 2 in our updated manuscript.
>
> **The reviewer asks about the potential of training our model using self-supervised objectives, such as those employed in DINO.**
>
> The reviewer raises an excellent point about the synergy between our architectural design and self-supervised pre-training schemes. As the reviewer correctly highlighted, DINO (and DINOv2) typically use ViTs. We believe our hierarchical design could be adapted by: (1) introducing a global CLS token in our final stage, where we use global attention, and (2) applying DINOv2's objectives to our final group embeddings, as opposed to patch embeddings used in ViTs. We believe that these modifications could further enhance the emergence of semantically meaningful object-centric representations in our final group tokens.
>
> Overall, we agree that this represents a compelling direction for follow-up works, and could unlock large-scale pre-training of native segmentation architectures.
>
>
> Lastly, we also thank the reviewer for noting the label overlap in Figure 5 of the supplementary material, which will be corrected in our updated manuscript.

---

### Official Review · Reviewer_dauE · 2025-07-01

**Clarity:** 4
**Significance:** 2
**Originality:** 3
**Rating:** 4
**Confidence:** 3

**Summary:**

This paper presents native group layer, enables tracking pixels from high-resolution during downsampling. The method can be seamlessly integrated with several segmentation tasks. The experiments demonstrate its effectiveness.

**Questions:**

(1) The description of M_loc in the algorithm section is too brief, yet this mechanism is arguably the core feature that distinguishes this method from query-based approaches. Based on the limited information provided, is it correct to assume that it simply masks out all regions that fall outside of a predefined rectangular window?

(2) Furthermore, since M_loc is non-learnable, this fixed masking rule might be unable to properly handle objects that span a very large area, such as a sky that occupies half of the screen. Does this local constraint pose a limitation for segmenting large-scale objects?

**Ethical Concerns:**

["NO or VERY MINOR ethics concerns only"]

**Final Justification:**

Thanks for authors' further explanation, which addressed my concerns. I keep my original scores.

**Limitations:**

(1) Given the current trend where open-vocabulary image segmentation is increasingly becoming a sub-task for Large Language Models (LLMs) and Large Vision Models (LVMs), I am curious about the potential for the native transformer architecture to be integrated with state-of-the-art VLMs, such as LISA.

(2) I understand that native segmentation appears to offer a different paradigm from the conventional segmentation head approach. However, I have reservations about its scaling potential. For instance, in Table 6, the model shows no performance advantage over Swin at the Large scale. Furthermore, Table 2 does not provide performance results for a "Large + native" configuration.

**Quality:**

3

**Strengths And Weaknesses:**

(1) It is interesting regarding the trackable downsampling of the Group layer, which is a potential way to balance segmentation quality (high-resolution) and memory costs.

(2) The paper is well-written, providing a comprehensive discussion with related works.

---

> ### Author Rebuttal · Authors · 2025-07-31
>
> We thank the reviewer for their positive assessment of our work, including recognizing our method's ability to perform segmentation efficiently, its effectiveness demonstrated through experiments, and the clarity of our writing. We are pleased to address their questions and concerns below.
>
> **The reviewer points out that the description on $M_{\text{loc}}$ in the algorithm section requires further elaboration.**
>
> We appreciate the reviewer's request for further clarification on $M_{\text{loc}}$. As the reviewer correctly points out, $M_{\text{loc}}$ indicates that input tokens only cross-attend over group tokens over a pre-defined rectangular patch, which we refer to as local grouping.  We note that $M_{\text{loc}}$ is used solely to illustrate that local grouping can be understood as a masking operation in the cross-attention-like operation introduced in Algorithm 1, line 3. In our implementation, as explained in Appendix E.1, $M_{\text{loc}}$ is not explicitly used, and the dot-product is only computed for non-masked pairs.
> We also note that we visually illustrate the behavior of $M_{\text{loc}}$ in Figure 2 (b), and further expand on it in Section 3.1 “Local and Dense Grouping”. In our revised manuscript, we will update the text to clarify the explicit connection between $M_{\text{loc}}$ further, as introduced in Algorithm 1, and the local grouping discussion to improve clarity.
>
> **The reviewer asks if $M_{\text{loc}}$ comes with potential limitations for segmenting large-scale objects.**
>
> The reviewer's question touches on a fundamental aspect of our grouping design. We would like to clarify that local grouping (formalized through $M_{\text{loc}}$) is only applied in the two intermediate grouping layers of our architecture. In the last grouping layer, we perform global (or dense) grouping without masking. Therefore, final groups can span large-scale objects without any constraint. The distinct behavior of different grouping stages is extensively visualized in Figures 4 and 5. We observe that our model’s output segments often represent entire background regions, such as ‘sky,’ spanning large regions. The reviewer is correct that if $M_{\text{loc}}$ were to be applied to our final grouping layer too, our model’s ability to segment large objects would have been compromised.
> In Table 3 a), row #4, we ablate quantitatively the role of disabling $M_{\text{loc}}$ (i.e., local grouping) in the last stage, and observe a drop of 2.5 mIoU on supervised fine-tuning on ADE20k, and 1.5 mIoU on zero-shot segmentation on Pascal VOC, confirming the expected behavior.
>
> **The reviewer asks how our method could be integrated into VLMs.**
>
> We thank the reviewer for pointing out this exciting research direction. We believe SeNaTra is particularly well-suited to be integrated into VLMs, and highlight two exciting directions:
> - Our output group tokens could enable an improved learnable image tokenization scheme, where visual tokens are better aligned with semantically meaningful concepts, by explicitly representing regions, objects, and their parts. It could further enable efficiency gains by reducing the number of redundant tokens in background regions (i.e., utilizing a single token to represent ‘sky’, instead of multiple rectangular patches).
> - For models such as LISA, which explicitly aim to obtain segmentations and use explicit mask decoders, our proposed native segmentation architecture could enable a more efficient design that could significantly reduce the burden on such mask decoders.
>
> Moreover, based on the remarkable performance obtained for zero-shot segmentation under CLIP-style text-supervision in Section 4.1.2, and the success of CLIP-based models as vision encoders in the context of VLMs, we hypothesize that SeNaTra could be a great candidate to be integrated into VLMs for future work.
>
> **The reviewer asks about the potential of our method under parameter scaling.**
>
> We appreciate the opportunity to clarify our parameter scaling results. Empirically, we observed that our model provides consistent improvements over Swin-Large; see Table 2. To be specific, in Table 2 (a), Ours-Large + M2F yields +0.6 mIoU for Semantic Segmentation, and in Table 2 (b), in the same setup, we observe +0.3 PQ.
> Below, we further provide results for Ours-Large+Native for both semantic segmentation on ADE20k and panoptic segmentation on COCO. As in Table 2, we compare our native results against two strong baselines: Swin-L+UperNet and Swin-Large + MaskFormer. All models are pretrained on ImageNet-22k and evaluated on the same input resolution:
>
> Semantic Segmentation on ADE20k
> | Backbone | Head        | mIoU | Params |
> |----------|-------------|------|--------|
> | Swin-L     | UperNet     | 52.1 | 234M   |
> | Swin-L     | MaskFormer  | 54.1 | 212M   |
> | Ours-L     | Native      | **54.7** | 211M   |
>
> Panoptic Segmentation on COCO-Panoptic
> | Backbone | Head        | PQ   | Params |
> |----------|-------------|------|--------|
> | Swin-L     | MaskFormer  | 52.7 | 212M   |
> | Ours-L     | Native      | **53.5** | 211M   |
>
> These results further demonstrate that our model consistently outperforms baselines under larger parameter regimes. We will include these comparisons in Table 2 in our revised manuscript.

---

> > ### Comment · Reviewer_dauE · 2025-08-04
> >
> > Thanks for the author's reply,  I think the selection of layers to apply M_loc is relatively tricky, relying on the human priors. While most of my concerns are addressed, I maintain the score inclined to accept.

---

> > > ### Author Response · Authors · 2025-08-05
> > > **On the selection of layers to apply M_loc**
> > >
> > > We thank the reviewer for their feedback on our rebuttal. We would like to clarify that the choice of applying M_loc (i.e., local grouping) to early stages of our architecture follows naturally from computational and architectural considerations:
> > > - In early stages the number of tokens is largest, making local grouping essential for scalability w.r.t image resolution.
> > > - Conversely, in the last stage, the number of tokens is smallest, and therefore global or dense grouping becomes computationally manageable and enables computing full masks efficiently.
> > >
> > > - As we show quantitatively in Table 9 (Appendix E.2), our design is crucial to enable scaling to large image resolutions. Our model’s memory requirements scale ~linearly with input resolution and can comfortably accommodate 1024x1024 images with <1GB of VRAM, comparable to a standard backbone, while a naive baseline not leveraging local grouping would require >40GB for the same resolution, making it impractical for downstream segmentation tasks.
> > >
> > > Overall, the general design of applying local grouping on all stages but the last one is motivated by the goal of enabling masks covering arbitrarily large regions while maintaining computational scalability. We therefore consider this to be a natural choice, exploiting problem constraints and yielding significant efficiency gains.

---

### Official Review · Reviewer_7tgC · 2025-07-02

**Clarity:** 3
**Significance:** 3
**Originality:** 3
**Rating:** 4
**Confidence:** 3

**Summary:**

This paper proposed native segmentation vision transformer. It uses the spatial grouping layer to replace uniform grid-based downsampling with learned dynamic assignment of visual tokens to semantically coherent groups based on image content. Successive grouping operations across backbone stages naturally compose into a mapping from input pixels to final tokens, effectively creating a multi-scale hierarchy of segmentation masks for tokens at each backbone stage. Experiments conducted on zero-shot segmentation tasks across multiple benchmarks demonstrate that the proposed model significantly outperforms prior art, including models trained on an order of magnitude larger datasets.

**Questions:**

1.	The grouping tokens are initialized via strided convolution. How impactful is this initialization approach compared to random or learned initializations as used in Slot Attention?
2.	Have the authors evaluated whether tokens from intermediate stages (e.g., stage 2 or 3) are semantically meaningful enough for classification or segmentation? Could this allow for controllable granularity or multi-scale segmentation?
3.	Given that grouping tends to merge spatially close tokens, does the model struggle with small object segmentation?
4.    How about the results compared to SAM2?

**Ethical Concerns:**

["NO or VERY MINOR ethics concerns only"]

**Final Justification:**

My concerns have been addressed.

**Limitations:**

Yes.

**Quality:**

3

**Strengths And Weaknesses:**

Strengths:

1.	The paper introduces the concept of "native segmentation," which generates high-quality segmentation masks solely through the backbone network without requiring additional segmentation heads. The spatial grouping layer is ingeniously designed, leveraging a differentiable iterative clustering algorithm, demonstrating strong innovation..
2.	The paper is well-written and easy to follow.
3.	The paper includes evaluations across classification, zero-shot, and supervised segmentation, alongside detailed ablations, which thoroughly validate the approach.
4.	The method shows strong performance in both zero-shot and supervised segmentation with fewer parameters and FLOPs than many baselines.

Weakness:

1.	The grouping tokens are initialized via strided convolution. How impactful is this initialization approach compared to random or learned initializations as used in Slot Attention?
2.	Have the authors evaluated whether tokens from intermediate stages (e.g., stage 2 or 3) are semantically meaningful enough for classification or segmentation? Could this allow for controllable granularity or multi-scale segmentation?
3.	Given that grouping tends to merge spatially close tokens, does the model struggle with small object segmentation?

---

> ### Author Rebuttal · Authors · 2025-07-31
>
> We are thrilled that the reviewer finds our paper well-written, our method well-performing and thoroughly evaluated, and our grouping layer to be a “strong innovation” and “ingeniously designed”. We are grateful for detailed comments and respond to detailed questions below.
>
> **The reviewer asks how the initialization strategy of group tokens compares to random or learned initializations.**
>
> We compare our initialization strategy to both randomly sampled embeddings (from Slot Attention [15]) and learned embeddings [63] in Section 4.3 and Table 3 (b). We empirically observed that random sampling led to training instability and divergence under standard learning rate configurations, and learned embeddings performed significantly worse (row #4 in Table 3 (b)). Our approach, utilizing strided convolutions, provides fast and input-dependent initialization with greater flexibility while remaining computationally efficient.
>
> We will expand on the existing discussion in Section 4.3. in our manuscript
>
> **The reviewer asks how semantically meaningful the tokens from intermediate stages are, and asks about their potential to enable multi-granularity or multi-scale segmentation.**
>
> In Figures 4 and 5 in the appendix, we observe that intermediate representations from stages 2 and 3 preserve semantic boundaries among objects and their parts, and can often correspond to fine-grained structures and small objects in the input (e.g., small fruits in Figure 4, row #1). However, due to the local grouping constraint applied in these stages, these intermediate tokens have limited receptive fields and therefore do not necessarily capture complete objects or semantic parts in their entirety.
>
> Overall, the segments from these intermediate stages provide boundary-preserving representations that facilitate progressive, hierarchical upsampling. As demonstrated in Table 3(a), employing grouping in intermediate layers significantly outperforms using a single dense grouping layer (row #2).
>
> Furthermore, the reviewer's insight regarding controllable granularity and multi-scale segmentation is well-founded. The hierarchical nature of our token representations is inherently suited for such applications. For example, in interactive annotation scenarios, intermediate-stage tokens could enable annotators to correct segmentation errors at various levels of detail, presenting a promising area for future research.
>
> **The reviewer asks how our method performs in the segmentation of small objects.**
>
> Our grouping algorithm incorporates both spatial proximity cues (through relative position encodings) and semantics via feature embeddings. Thus, it has all the necessary information needed to segment small objects.
>
> We quantify this observation by computing Average Precision (AP) for our panoptic segmentation models separately over different object sizes (Large, Medium, and Small, as determined by the original COCO tools package), and report results compared to baselines for both native and Mask2Former (M2F) based results below:
>
> | Backbone | Head   | PQ   | AP   | AP$^{\text{large}}$ | AP$^{\text{medium}}$ | AP$^{\text{small}}$ |
> |----------|--------|------|------|----------------------|-----------------------|----------------------|
> | Swin-T    | MF     | 47.7 | 33.6 | 55.8                 | 38.6                  | 15.2                 |
> | Ours-T     | Native | **49.2** | **38.2** | **60.3**                 | **42.1**                  | **18.2**                 |
> | Swin-T     | M2F    | 53.2 | 43.3 | 65.4                 | 47.0                  | 23.6                 |
> | Ours-T     | M2F    | **55.0** | **44.9** | **66.9**                 | **49.1**                  | **25.6**                 |
>
> As shown in the table, our method demonstrates strong performance on small object segmentation, achieving absolute improvements of +3.0 AP$^{\text{small}}$ in the native setting (18.2 vs 15.2) and +2.0 AP$^{\text{small}}$ with M2F heads (25.6 vs 23.6) compared to Swin baselines.
>
> As can be seen, our method outperforms significantly both methods in terms of AP$^{\text{small}}$, showing its ability to segment objects of small scale. We will add these results to the appendix.
>
> **The reviewer asks how our method compares to the Segment Anything (SAM) model.**
>
> We appreciate the reviewer's important question regarding the comparison with SAM and SAM2. We would like to clarify the distinct focus of our work. SAM and SAM2 introduce a comprehensive framework comprising three key components: the promptable segmentation task; the associated dataset and its generation engine; and a segmentation model, which employs a conventional ViT/Hiera backbone image encoder.
>
> In contrast, our work focuses on the development of novel image segmentation backbone architectures that can replace conventional encoder/decoder networks for training image (segmentation) foundation models, such as SAM.
>
> Our model’s core contribution, the spatial grouping layer, allows segmentation masks to emerge directly from the backbone, enabling segmentation without the need for dedicated decoder heads. This native segmentation capability could streamline SAM's model architecture, potentially offering a more efficient decoder design. This may lead to better scaling properties and comparable performance while requiring fewer annotations, as demonstrated by our image-text pre-training results in Section 4.1.2; however, such scaling efforts are out of the scope of this paper.  We will elaborate on this distinction in the revised version of our paper.

---

### Decision · Program_Chairs · 2025-09-17

**Decision:**

Accept (poster)

**Comment:**

Four reviewers recommend to accept (3 borderline) the paper. The main reasons for acceptance are the novel architecture for segmentation, the strong results (especially in zero-shot settings), and the potential for impact in research and practical applications. The main weaknesses are relatively minor and relate to the breadth of experimental comparisons, ablation studies, and some open questions about scaling and integration with other paradigms. The reviewers agree that the paper is well-written, innovative, and makes a valuable contribution.
On balance, the AC sees not basis to overturn the reviewer suggestions and recommends acceptance. The paper presents an interesting approach to native segmentation in vision transformers, with strong empirical validation and good potential for future impact. The AC highly recommends the authors to address the concerns of the reviewers and take into account their suggestions of improvement when preparing a revised version.